# Microorganisms Associated with Mosquito Oviposition Sites: Implications for Habitat Selection and Insect Life Histories

**DOI:** 10.3390/microorganisms9081589

**Published:** 2021-07-26

**Authors:** Maxime Girard, Edwige Martin, Laurent Vallon, Vincent Raquin, Christophe Bellet, Yves Rozier, Emmanuel Desouhant, Anne-Emmanuelle Hay, Patricia Luis, Claire Valiente Moro, Guillaume Minard

**Affiliations:** 1Univ Lyon, Université Claude Bernard Lyon 1, CNRS, INRAE, VetAgro Sup, UMR Ecologie Microbienne, F-69622 Villeurbanne, France; maxime.girard@etu.univ-lyon1.fr (M.G.); edwige.martin@univ-lyon1.fr (E.M.); laurent.vallon@univ-lyon1.fr (L.V.); vincent.raquin@univ-lyon1.fr (V.R.); hay.de-bettignies@univ-lyon1.fr (A.-E.H.); patricia.luis@univ-lyon1.fr (P.L.); claire.valiente-moro@adm.univ-lyon1.fr (C.V.M.); 2Entente Interdépartementale Rhône-Alpes pour la Démoustication, F-73310 Chindrieux, France; base.decines@eid-rhonealpes.com (C.B.); yrozier@eid-rhonealpes.com (Y.R.); 3Univ. Lyon, Université Claude Bernard Lyon 1, Laboratoire de Biométrie et de Biologie Evolutive, UMR CNRS 5558, VetAgro Sup, F-69622 Villeurbanne, France; emmanuel.desouhant@univ-lyon1.fr

**Keywords:** microbiota, microbiome, mosquitoes, behavior, oviposition, larval habitat, life history traits, nutrition, development, survival

## Abstract

Mosquitoes are considered one of the most important threats worldwide due to their ability to vector pathogens. They are responsible for the transmission of major pathogens such as malaria, dengue, zika, or chikungunya. Due to the lack of treatments or prophylaxis against many of the transmitted pathogens and an increasing prevalence of mosquito resistance to insecticides and drugs available, alternative strategies are now being explored. Some of these involve the use of microorganisms as promising agent to limit the fitness of mosquitoes, attract or repel them, and decrease the replication and transmission of pathogenic agents. In recent years, the importance of microorganisms colonizing the habitat of mosquitoes has particularly been investigated since they appeared to play major roles in their development and diseases transmission. In this issue, we will synthesize researches investigating how microorganisms present within water habitats may influence breeding site selection and oviposition strategies of gravid mosquito females. We will also highlight the impact of such microbes on the fate of females’ progeny during their immature stages with a specific focus on egg hatching, development rate, and larvae or pupae survival.

## 1. Introduction

Animals often recognize microbes in their environment due to the simple fate that they can smell or taste microbial compounds and use that information to adapt their behavioral response to the environmental conditions [1]. Firstly, microbial cues from an animal microbiota may attract conspecifics. As an example, aggregation of the desert locust *Schistocerca gregaria* is mediated by Guaiacol, a pheromone derived from lignans metabolism by the locust bacteriome then released in adult feces [2,3,4]. Secondly, animals can use pathogens’ cues in species recognition and avoid unfavorable interactions. Thus, rodents show reduced motivation to engage in social interactions with sick conspecifics after smelling odors characteristic of bacterial infection [5]. Thirdly, microbes can be involved in interindividual communications. Bacterial ketones and alcohols released in the urine of the African elephant male inform others about their musth status (i.e., an aggressive behavior induced by a major shift in their hormonal balance) [6,7]. Finally, microbial cues can be used to locate suitable food sources and habitats. The fruit fly *Drosophila melanogaster* avoids geosmin, a volatile released by *Penicillium* fungi and *Streptomyces* bacteria, which grow on decaying fruits and that are lethal to the insect [8]. Basic knowledge of the influence of microbes on animals offers innovative strategies to control invasive species, pests, and vectors.

Amongst arthropods, mosquitoes (Diptera: Culicidae) form a highly diversified family with more than 3601 different species divided into two different sub-families: Anophelinae (482 species) and Culicinae (3119 species) [9]. Mosquitoes are the major disease vectors worldwide with some species being able to transmit pathogens of public and veterinary importance. For example, *Aedes* mosquitoes transmit arboviruses including dengue, chikungunya, and yellow fever viruses while *Anopheles* are the vectors of *Plasmodium* spp. parasites responsible for malaria [10]. Several physiological, ecological, and environmental factors impact the probability of mosquitoes to transmit pathogens in the field such as (i) vector density and biting rates, (ii) pathogen survival, (iii) host-vector contact as well as (iv) insect vector competence. The latter is defined as the ability of pathogens to efficiently colonize the vector, to replicate and get transmitted under controlled conditions [11]. Therefore, limiting the density of vector populations below the transmission threshold (i.e., the critical level of vector density above which the introduction of a few infectious individuals into a community of susceptible individuals will give rise to an outbreak) is a keystone action that can be performed in order to limit the expansion of mosquito-borne diseases. To that end, methods mainly based on the use of chemical insecticides have been applied to control mosquitoes. As an example, the Center for Disease Control and Prevention (Atlanta, GA, USA) recommends their use inside housing in order to limit malaria transmission. Such a strategy has led to a 21% decrease of malaria cases over the world between 2012 and 2015 [10]. Despite their proven efficiency, chemical insecticides often (i) lack specificity and impact on untargeted species, (ii) led to the selection of mosquito resistant populations as previously evidenced for dichlorodiphenyltrichloroethane (DDT) and pyrethroids, (iii) led to health issues, in particular when they are used indoor [12,13,14]. To overcome these undesirable effects, alternative strategies have gradually been developed. Among them, insect-chemoattractant/repellent compounds as well as organic insecticides, most often originating from microorganisms, have been applied in the field [15,16,17,18,19].

Mosquitoes are holometabolous insects meaning that they will proceed to a complete metamorphosis. After the egg has hatched in aquatic environment, individuals will follow a post-embryonic development starting with a larval stage and a pupal stage to finally emerge as an imago. Each stage but imago colonizes aquatic habitats. Larvae use different feeding strategies such as filtering, suspension feeding, grazing, interfacial feeding, or predation, to acquire organic matters within their aquatic habitats [20]. They developed into four different instars that are separated by exuviations and metamorphose into pupae before emerging as an adult at the interface between air and water. After being mated by males, females of anautogenous species (most species such as *Aedes albopictus*) will bite a vertebrate host in order to acquire essential amino acids required for egg maturation [21]. Conversely, autogenous species (*Malaya* spp., *Toxorynchites* spp., and *Topomyia* spp.) can lay eggs without ingesting any blood meal. Recognition and selection of breeding sites by gravid females is a key step in mosquito life cycles. Since a single mosquito female lays multiple clutches during its whole life and since each clutch is ranging from tens to hundreds of eggs without no parental care, it is of primary importance to manage larval habitats. For instance, *An. gambiae* females can delay egg laying up to 50 days in absence of suitable breeding sites [22]. This drastically impacts the fitness of individuals by reducing egg hatching and larval development rates. Even if all mosquitoes are selecting aquatic habitats, each species search for and select certain characteristics of these habitats (e.g., in term of salinity, sunlight exposition, stream flow, type of predators…) [21,23]. As an example, the mosquito species *Aedes taeniorhynchus* and *Anopheles crucians* tend to prefer domestic habitats and lay eggs in artificial containers while other species such as *Culiseta melanura* prefer sylvatic sites and freshwater swamps [24].

Egg laying site selection is a keystone behavior determining the fate of the female progeny and, thus, is expected to be under strong selective pressures. Such localization and selection of water habitats by gravid females involve olfactory, visual, gustatory, and tactile signals [25]. Mosquitoes detect olfactory signals with their antennae, maxillary palps, and proboscis [26]. Tarsal segments of the legs, the labellum and labrum of the mouthparts, and the cibarium, an internal organ, are rather important for tasting and sensing the breeding site [27]. These organs contain multiporous sensory hairs called sensilla that house olfactory sensory neurons expressing chemosensory receptors that are detecting specific compounds. Phenotypic responses of gravid females to environmental signals might vary. Some signals can be classified as (i) “attractant” if they elicit insect-oriented movement toward the source, (ii) “repellent” if they induce insect-oriented movement away from the source, (iii) “stimulant” if they elicit oviposition, and (iv) “deterrent” if they prevent oviposition (Figure 1; [28]). Those water habitats are colonized by a wide variety of prokaryotic and eukaryotic microorganisms. Due to their ability to synthesize compounds with organoleptic properties, they have been shown to influence the mosquito oviposition site selection.

In this review, we synthesize current knowledge on the influence of exogenous microorganisms colonizing larval habitats both on the oviposition strategy of gravid female mosquitoes and on their offspring performance in terms of development and survival. Small crustacean (e.g., copepods) will not be treated here but have previously been reviewed [29].

## 2. Influence of Microorganisms on the Mosquito Oviposition Site Selection

Mosquitoes water habitats are often rich organic matter acquired from soil, vegetation, animal cadavers, and dejections [20]. Such microenvironments promote the growth of a wide variety of microorganisms, which have been shown to be key drivers for communities assembly of mosquitoes microbiota and determine major adult traits [30,31,32,33,34,35]. In this section, we review how microorganisms’ cues either attract/stimulate or repel/deter gravid females. We sum up the knowledge about the characteristics of these microbial kairomones (i.e., semiochemical compounds that are produced by microorganisms and recognized by mosquitoes) and discuss how variations in microbial densities might elicit drastically contrasted behavioral responses in mosquitoes.

### 2.1. Do Microorganisms from Water Habitats Attract/Stimulate Gravid Females?

Plants have been shown to attract gravid females for oviposition [36]. However, a recent study evidenced that plant associated microbes might be responsible for some observed behavioral responses. Using dual choice experiments, Arbaoui et al., [37] demonstrated that the average number of eggs laid by the yellow fever mosquito *Ae. aegypti* is 20 times higher in bamboo infusions from *Bambusa* spp. than in distilled water. The authors attempted to determine whether microbial cells were mandatory to elicit this effect by filtrating them with 0.45 µm pores beforehand; however, they did not observe any differences between the number of eggs deposited on filtrated or non-filtrated infusions. This suggests that microbial cells (that are usually larger than 0.45 µm) are not directly responsible for this response. Instead, compounds might rather be the mediators (eventually secreted from microbes). Similarly, another study showed that females laid more eggs on infusions from the bamboo species *Arundinaria gigantea* or a leaf infusion from white oak than on distilled water [38]. Both studies suggest that plant infusions promote the oviposition behavior of gravid female mosquitoes; microbes associated with those infusions could provide information through chemical compounds. A total of 14 and 18 bacterial isolates were cultivated from the infusions of the bamboo A. gigantea and white oak, respectively. An important density of Alpha and Gammaproteobacteria were retrieved. A purified blend of organoleptic carboxylic acids synthesized by those bacteria recapitulates the oviposition behavior observed with bamboo infusion, with tetradecanoic acid being identified as the major attractive compound. A total of 10 individual isolates belonging to the genus *Bacillus, Enterobacter. Pseudomonas, Lactococcus, Enterobacter*. *Shigella, Citrobacter, Brevundimonas* and six individual isolates belonging to the genus *Bacillus, Lactococcus, Enterobacter, Pseudomonas, Citrobacter, Brevundimonas* were still attracting/stimulating Ae. aegypti and Ae. Albopictus, respectively, while used individually at intermediate concentrations and compared to water. On the other hand, lower concentrations did not attract/stimulate mosquitoes and higher concentrations repel/deter them.

Even if plant infusion and their associated microorganisms were shown to be good elicitors of mosquitoes’ oviposition, natural breeding sites often contained a more variable diversity and abundance of microorganisms. Therefore, to mimic natural conditions, other authors tested the effect of water from natural oviposition sites without *a priori* on the nature of water [39]. They showed that fresh soils or water collected in known oviposition sites of the malaria vector *An. gambiae* received, respectively, 3.9 and 2.6 times more eggs than sterile distilled water when the choice was offered to gravid females in dual choice experiments. To ensure that only olfaction rather than touching and tasting could be involved in the recognition, the authors used an experimental system preventing female mosquitoes from touching the substrate. Similar results were obtained using sterilized substrate (autoclaved soil of filtered water) instead of sterile water. Isolated bacteria (including unclassified Firmicutes, *Aeromonas, Pasteurella, Pseudomonas, Vibrio, Acinetobacter*, and Enterobacteriaceae species) from soil collected beneath oviposition sites and larval habitats restored the attractiveness/stimulant properties of sterile soils but not filtered distilled water. These results suggest that the dilution of microorganisms or volatile organic compounds (VOC) into water might decrease the capacity of mosquitoes to use kairomones as an information source. Volatiles from bacteria isolated in this experiment were then analyzed [40]. It appeared that the bacteria correlated with a positive oviposition response clustered into different groups. The authors suggest that different molecules produced by those bacteria and recognized by the mosquito might differ across bacterial isolates. When combined with previous results obtained from mosquito antennae electro-physiological response studies toward volatiles, a list of potential attractive compounds was updated and restricted to aliphatic alcohols (2-Methyl-3-decanol, methyl-1-butanol), aromatic alcohols (2-phenylethanol, phenylmethanol), indole, pyrazines (alkyl-pyrazines), and carboxylic acids (3-methylbutanoic acid). More recently, lake water supplemented with six days-old soil infusions from breeding sites was shown to efficiently attract gravid *Anopheles gambiae s.l*. females [41]. However, this attractiveness disappeared after autoclaving the mixture. The authors characterized cedrol, a sesquiterpene alcohol, as a major attractant present in the infusion and showed that natural habitats in which cedrol was identified were more likely to be colonized by *Anopheles* mosquitoes [42]. Finally, they identified two endophytic fungi (a species of the *Fusarium fujikuroi* complex and *F. falciforme*) from rhizomes in soils beneath *Anopheles* oviposition sites, able to produce cedrol and some of its analogues [43]. This set of results represents major advances in the identification of the molecules or blend that attract female mosquitoes. However, the list is certainly far from exhaustive. Indeed, field surveys often reported that many presumably suitable breeding sites for *Anopheles* mosquitoes remained uncolonized [44,45,46]. Those observations suggest that important factors influencing breeding site selection might be missing to predict the attractiveness and potential suitability of those habitats.

Apart from environmental microbes, parasites and symbionts might also be involved in mosquito orientation and stimulation toward specific breeding sites. *Ascogregarina* spp. are eukaryotic parasites that specifically colonize *Aedes* and *Culex* mosquitoes [47]. The species *As. taiwannensis* shows attracting properties toward *Ae. aegypti* (one of the closest species of its natural host *Ae. albopictus*) [48]. On average, females lay 3.6 times more eggs in water containers supplemented with parasitized larvae than in containers supplemented with non-parasitized larvae, larvae parasitized with an entomopathogenic fungi (*Smittium morbosum*), or containers that were not supplemented with larvae. *As. taiwanensis* is not able to complete its life cycle when colonizing *Ae. aegypti*, therefore, it might be possible that the attractiveness underlying mechanism is generalist and could attract both this neighboring species and its natural host *Ae. albopictus*. However, this latter property has never been tested until now. The same authors also demonstrated that *Candidatus* near *pseudoglaebosa*, a yeast symbiont colonizing the gut and the peritrophic matrix of *Ae. aegypti* is also more acceptable to ovipositing females than distilled water or rearing water from uninfected larvae [48]. Those results suggest that some microorganisms have evolved strategies that specifically attract gravid females and consequently maximize their chance to encounter mosquitoes and in turn benefit from that interaction to accomplish their lifecycles and/or get dispersed.

Many studies reported the influence of microorganisms on oviposition elicitation. A summary of attractive/stimulating microorganisms involved in mosquito oviposition site selection is detailed in Table 1. However, most of those studies cannot discriminate attractant from stimulant effect [25] and terminology should be taken cautiously. The experimental design of current studies is often based on (i) dual choice experiment that only shows attractiveness or (ii) the number of eggs laid when different habitats are proposed, which can both consist in an attractiveness or a stimulation. However, the sole effect of stimulation by microorganisms has poorly been addressed. Stimulant bacteria might increase both the number and frequency of egg laying without modifying the mosquito choice. Influence of those effects should be considered for future prospect given their potential consequences on mosquito population dynamics.

### 2.2. Do Microorganisms Repel/Deter Gravid Females?

The microsporidian parasite *Edhazardia aedis* is an intracellular obligate parasite that specifically infects the mosquito *Ae. aegypti* [57]. This parasite strongly affects the survival and reproductive success of the mosquito [58]. Its life cycle is complex since the microsporidian spores can be both vertically and horizontally transmitted with a high transmission success [59]. Due to its high transmission rate and maintenance in mosquito populations, the parasite was proposed as a promising candidate for mosquito biological control [60,61]. However, the ability of uninfected *Ae. aegypti* females to avoid egg deposition when oviposition sites are colonized by infected conspecific larvae questions its use [54]. Indeed, dual choice experiments demonstrated that uninfected females laid a higher proportion of eggs (60.8 ± 2.1%) in cups containing uninfected larvae. The potential semiochemicals involved in attractiveness differentiation were not identified to date. Such a strategy might be an evolutionary response of the mosquito toward the fitness cost of the parasite in natural populations. However, the oviposition deterrence is not complete, which also suggests that in the field, a part of the population will get infected, enabling the parasite to complete its lifecycle and spread among individuals when a part of the population remains uninfected. The trematode *Plagiorchis elegans* is another parasite of *Ae. aegypti*. The presence of this parasite in the water, or in a snail host living in aquatic habitats, does not seem to affect the oviposition behavior of gravid females [55]. However, as previously described for *E. aedis* dual choice experiments showed that breeding sites containing infected larvae were repellent/deterrent toward gravid females and accumulated fewer eggs than sites containing uninfected larvae or solely water [55,56]. This repellent/deterrent effect was still observed when water was treated with antibiotics or boiled, suggesting that (i) presence of the parasite was not mandatory and (ii) that thermostable non-volatile compounds have been produced by infected larvae or by the parasite to mediate breeding site recognition by mosquitoes. In addition, the repellent/deterrent effect was increased when water was filter sterilized, with 10 times more eggs in containers with uninfected larvae. This difference was attributed to bacteria colonizing the containers, such as *Flavobacteria* sp., that attract mosquitoes, thus, mitigated the repellency of the parasite. Contrarily to this previous experiment where water was regularly changed, a recent study conducted with water that was not changed for 14 days and potentially accumulated bacteria, failed to observe the repellent/deterrent effect of *P. elegans* infected mosquitoes [62]. This confirms that, due to presence of bacteria in water containers, repellency/deterrence of the parasite might often be mitigated and has rarely been observed in the field. Since *Ae. aegypti* lay eggs in standing water, it may be possible that the potential repellency/deterrent effect of *P. elegans* would not be efficient in the field. *Bacillus thuringiensis* var. *israelensis* (Bti) is a dipteran pathogen that has been broadly used in biological control against *Aedes, Culex,* or *Anopheles* mosquitoes [63]. Depending on the species, female mosquitoes do not respond similarly to the presence of Bti in water habitats. Indeed, *Culex quinquefasciatus* tend to lay less eggs in Bti-infected water containers compared to sterile water [50]. In addition, the number of eggs laid as well as the size of egg rafts negatively correlated with the concentration of Bti. On the opposite, no influence of Bti was observed toward *An. arabiensis* female behavior [51] and from no effect to a slight attractive/stimulant effect was even reported for *Ae. albopictus* [52,53]. Those differences might be explained by the fact that *Culex* mosquitoes drink water before laying eggs and might recognize solubilized compounds with their phagoreceptors as previously discussed [54]. However, those conclusions should be taken cautiously because different dose of Bti were used in those experiments and mosquito species effects might be confounded with dose effects, which could have also led to differences in gravid female responses. A summary of repellent/deterrent microorganisms involved in mosquito oviposition site selection is detailed in Table 1.

As well as what was reported for attractant/stimulant effects, many studies cannot discriminate repellent from deterrent effects [25] and deterrence has poorly been regarded. Further prospect might be necessary to broaden our knowledge on the influence of microbes on such behavior. In addition, the influence of microorganisms on either attraction/stimulation and repellency/deterrence is highly influenced by the density of cells and signals within the breeding sites.

### 2.3. How Dose-Response Effects Influence the Female Oviposition?

A single microorganism can both attract/stimulate or repel/deter mosquitoes depending on its concentration within a breeding site. Exposure to a bacterial mixture of 10^6^, 10^7^, and 10^8^ cells/mL attracts female mosquitoes of the species *Ae. albopictus* and *Ae aegypti*, while higher concentrations of 10^9^ cells/mL had no effect on the former species and even repel the latter one [49]. When tested individually, seven of those bacterial isolates (*Bacillus thuringiensis, Enterobacter asburiae, Enterobacter cancerogenus, Lactococcus lactis, Shigella dysenteriae, Citrobacter freundii*, and *Brevundimonas vesicularis*) attracted *Ae. aegypti* at only two out of three concentrations, while two of them attracted it at one concentration and one of them repelled it at one concentration. In comparison, *Ae. albopictus* mosquitoes showed mitigate responses since three single isolates showed a drastically different response at each of the three concentrations. The most noticeable one was *Brevundimonas vesicularis* that strongly attracts females at 10^6^ cells/mL, does not influence them at 10^7^ cells/mL, and strongly repel them at 10^8^ cells/*mL*. Variation in secondary metabolites concentrations has been proposed as a keystone effector leading to such differential behavioral responses. As an example, *Mycobacterium ulcerans* produces a toxin named mycolactone. When dual choice experiments were carried out with non-inoculated vs. inoculated containers, 64% of eggs were laid in containers with 1 µg/mL of toxin versus 38.3% and 41.6% in those supplemented with 0.5 and 0.05 µg/mL of toxin, respectively [64].

Those results point out that variation in microbial communities’ composition and density shape mosquito oviposition behavior by impacting the diversity and concentration of volatile compounds to either influence the behavior of gravid females. Therefore, identifying the volatile molecules and their dynamics in natural oviposition sites could be key to improve vector control strategies.

## 3. Influence of Microorganisms Colonizing Water Habitats on Mosquitoes’ Premature Life History Traits

One common evolutionary theory is that females’ preferences in oviposition sites are oriented to maximize the performance of their offspring since natural selection might filter-out the progeny from mothers that make the wrong decision. This point has previously been discussed [65] and several factors in mosquitoes’ ecology strengthen the prediction of a tight preference–performance coupling. First, mosquitoes are gregarious during the immature stages, which means that unsuitable habitat will have drastic consequences in terms of effective loss. Secondly, they cannot move from one habitat to another during immature stages, meaning that they cannot escape unsuitable habitats. Conversely, other characteristics argue against a preference–performance prediction. This concerns mosquito species laying large and sparse egg clutches. Contrarily to species that lay all their eggs in the same container, the consequences of selecting an unsuitable habitat might be diluted for these species across the progeny. Furthermore, environmental variations influencing the unpredictability of habitat quality during the oviposition period might also alter the importance of the females’ choice. Indeed, environmental stochasticity tend to select females that lay more eggs and/or in multiple habitats since the survival of their progeny is uncertain.

If the importance of microorganisms in the performance–preference coupling has been poorly addressed, several studies previously demonstrated that microbes colonizing water habitats influence the life history traits of mosquitoes with, even, drastic consequences on adult traits (see an example here [32]). In this section, we will more specifically comment the impact of microorganisms on larval nutrition, mosquito development (including egg hatching and post-embryonic development), and immature (eggs and larvae) survival.

### 3.1. Can Microorganisms Be Used as a Food Source by Mosquito Larvae?

Larvae acquire their organic matter from a wide diversity of food sources, which are often partly composed of plant or animal tissues as well as microorganisms that are harvested from the breeding sites. Thus, in nine different sites from Belize, the occurrence of *Anopheles albimatus* in marshes correlated with the presence of cyanobacteria [66]. Larvae were only observed in marshes that harbored mats characterizing the presence of Cyanobacteria. The cyanobacteria are supposed to modify habitat towards conditions more favorable to mosquito larvae (i.e., by elevating the temperature of water and producing more CO_2_ that would be recognized by females as an attractant). It has been also suggested that cyanobacteria themselves might constitute a suitable food source for mosquito larvae [67]. Indeed, when the cyanobacteria *Phormidium animalis* was cultured in the presence of *An. albimanus* larvae, (i) larvae were able to ingest cyanobacterial cells, (ii) cells were identified in larval guts within a 30 min laps time, and (iii) filamentous cells were partially digested in 180 min. Beyond this time, it was not possible to cultivate *P. animalis* suggesting that the cyanobacteria were killed when they passed through the gut. Vázquez-Martínez et al. also demonstrated that larvae successfully developed without any additional source food. When compared with conventional rearing methods (i.e., feeding larvae with germinated wheat), the larval development and emergence rate were slightly lower in the presence of cyanobacteria, but the size of adults (wing length) and sex ratio were equivalent for both treatments.

Apart from Cyanobacteria, other microorganisms have been shown to influence mosquito nutrition. *Culex* species prefer to lay eggs in water containing yeast extracts compared to sterile water [68]. Given this choice, one can hypothesize that females select their breeding sites in order to favor the development of their offspring and, therefore, that yeast extracts could be involved in such a trait. Larvae reared with an optimal food source (i.e., fish food) or with *S. cerevisiae* exhibited a similar development rate, while those fed with various bacterial species as a sole food source developed slower [69]. In another study, several microorganisms including yeasts, bacteria, and algae were used as a sole food source and their effect on development were compared in *Ae. aegypti* larvae [70]. Digestion of those microorganisms by the larvae was tested by following the transfer of fluorescence from labelled microorganisms to the larvae. The authors demonstrated that yeasts had higher carbohydrate and protein contents than bacteria. Overall, the yeasts *S. cerevisiae* and *Pseudozyma* sp. as well as the proteobacteria *Asaia* sp. and *Escherichia coli* could be used as nutrients by larvae while other microorganisms including cyanobacteria (*Arthrospira platensis*), firmicutes (*Bacillus* sp., *Staphylococcus aureus*), proteobacteria (*Ochrobactrum intermedium*), and the algae (*Chlorella* sp.) constituted a poor food source associated with low development rate.

If nutrient acquired from digested microbes can increase larval growth non-digested microbes have also been shown to influence the mosquitoes’ development.

### 3.2. What Is the Influence of Microorganisms on Mosquitoes’ Development?

#### 3.2.1. Egg Hatching

Mosquitoes often lay eggs at the surface of water plans or at the interface between water and containers. Egg hatching depends on several environmental stimuli including water vibration [71], agitation [72], and temperature [73]. Mosquitoes of the genera *Ochlerotatus* and *Aedes* usually respond to a decline in oxygen concentration after the eggs get immerged within water [74,75,76]. This decline has been attributed to bacterial bloom that are induced by an influx of nutrients carried by water during rainfall. Eggs of *Ae. aegypti* were exposed during five consecutive days to an oak infusion or to the same infusion filtered through a 0.22 µm pore membrane [77]. The hatching proportion of eggs exposed to the non-filtrated infusion was 94% while it was only 4% with the filtrated one. Eggs were then exposed to a mixture of 14 bacterial strains isolated from a bamboo infusion. Results showed that the egg hatching was proportional to the bacterial cell concentrations (i.e., 10^6^, 10^7^, 10^8^). Previous studies suggested that a slow deoxygenation of the water is a hatching stimulus for many mosquito species [78,79,80] suggesting that a consumption of oxygen by the bacteria might be responsible for the observed response. However, the comparison of hatching rate in water with a high O_2_ concentration supplemented with bacteria or sterilized by filtration indicates, regardless of O_2_ levels, that the bacteria themselves or their metabolites could stimulate *Ae. aegypti* egg hatching. Apart from commensal microbes, pathogens can also induce egg hatching to benefit their lifecycle. The entomopathogenic fungus *Tolypocladium cylindrosoporum* was often retrieved from *Aedes* sylvatic breeding sites in South America [81]. Aedine mosquito eggs are resistant to desiccation and will often hatch when they are flooded. However, previous studies suggest that *T. cylindrosoporum* is able to induce a premature hatching of *Ae. aegypti* and *Ae. albopictus* eggs under suboptimal conditions. Indeed, when reared at 15 °C, eggs hatched preferentially after 14 days without being flooded when incubated with the fungi while non treated ones hatched preferentially after 21 days. Flor-Weiler et al. suggest that premature egg hatching might benefit this entomopathogen to rapidly infect its larval hosts. However, those laboratory-made observations failed to get replicated in the field. The authors suggested that environmental variables might not be appropriate to induce a proper fungi filamentation under natural conditions and that infections might be limited to cold periods. A more recent study performed with two strains of *T. cylindrosporum* failed to reproduce those results. The authors did not observe any premature eclosion or malformation of the embryonic larvae within the eggs after their exposition to a concentration gradient of conidia [82].

All in all, the current literature shows that microorganisms play an important role in the oxygen signals determining egg hatching but other microorganism mediated stimuli should be further investigated.

#### 3.2.2. Post-Embryonic Development

Recent investigations showed that colonization of mosquitoes from the species *Ae. aegypti, Georgecraigius atropalpus, Toxorhynchites amboinensis*, and *An. gambiae* by microorganisms acquired from the water of larval habitat strongly influence the development of larvae [30,83,84]. Indeed, axenic larvae (i.e., deprived from microorganisms) exhibited either an absence of development after the first instar or a strong delay. Recolonization by multiple bacteria (including the non-symbiotic bacterium *Escherichia coli*) or fungi previously isolated from mosquitoes restored a complete development until the adult stage. When the development of axenic larvae was measured in presence of various *E. coli* mutants, cytochrome bd oxydase genes were proved to be essential to restore the development of mosquitoes [85]. These genes are involved in the microaerobic respiration of bacteria and further investigations suggested that consumption of oxygen by bacteria within the mosquito gut induces hypoxia, which leads to the expression of the Hypoxia Induced Factor-α (HIF- α). This factor in turn enhances the transcription of many genes involved in various steps of the ecdysis including gut growth, neutral lipid transport, and 20-hydoxyecdysone secretion (the major hormone involved in the signaling of ecdysis) [86]. More recent advances demonstrated that riboflavin is the mediator of this interaction since axenic larvae can actually develop into adult when reared under conditions that preserve this vitamin and exhibit an anoxic midgut [84,87]. Under natural conditions, this compound is provided by the larval microbiota. Valzania et al. showed that those hypoxia-related pathways might also be enhanced by eukaryotic microorganisms including the non-symbiotic algae *Chlamydomonas reidnartii* and the yeast *Saccharomyces cerevisiae* [88]. Another investigation on gnotobiotic mosquitoes also evidenced differences in the mosquito development rate when colonized by prevalent bacterial taxa isolated from natural breeding sites in Gabon [32]. This impact was also correlated with differences in pupation rates as well as carry over effects on adult mosquitoes including differences in size, immune response, and vector competence for dengue viruses.

Those results highlighted the major influence of microorganisms on the signal leading to larval development with consequences on the adult traits. However, most of those effects are not specific enough and further studies are necessary to determine to which extent microbial composition and density modulate the development of larvae.

### 3.3. Do microbes Impact the Post-Embryonic Survival?

The survival of immature stages of *Culex pipiens* has been shown to be impacted by yeast they feed on within habitat with some variations. Indeed, larvae exposed to *Metschnikowia bicuspidata* and *Wickerhamomyces anomalus* yeasts as a sole food source showed 70–80% of larval survival and 10–15% pupation, while those exposed to *Cryptococcus gattii* showed less than 30% larval survival and no pupation [89]. The soil-borne entomopathogenic filamentous fungi *Metharizium* sp. has been proposed as a potential biological control agent to fight mosquitoes. Spores of this fungus preferentially colonize terrestrial insects but have been reported to accidentally infect larvae of *Aedes, Anopheles*, and *Culex* mosquitoes [90,91] when dispersed in water [92]. The species *M. pingshaense* has recently been genetically modified to produce an insect-specific toxin and has been shown to be more efficient and specific than commonly used chemical insecticides to fight *Anopheles* mosquitoes [93]. Indeed, during lab trials, 75% of the females were reported as infected and survived on average five days after infection with a threefold reduction in the number of eggs laid. Densoviruses are part of the Parvoviridae family. They specifically colonize the nuclei of invertebrates’ cells and form large inclusion [94]. Mosquito densovirus might play an important role in mosquito population regulation and particularly during immature stages since larval infections often result in individuals’ death and major malformations [95]. Some microorganisms affecting larval mosquito survival have been broadly studied. The entomopathogenic bacteria Bti was successfully applied as biological agent to control mosquito larvae and commercially formulated to be dispersed within their water habitats since its discovery in 1976 [96]. This biological insecticide specifically affects the survival of Diptera and poorly impacts non-target species, which are often distant from this order (with some exceptions such as chironomids) [97]. The larvicidal effect of the Bti strains relies on the production or three peptidic Cry toxins that binds a cadherin receptor [98] within the gut epithelium of the mosquito digestive tract and form a cation channel. They work synergistically with Cyt toxins that acts as a detergent-like membrane perforator [63].

Studies are often largely biased toward microorganisms that would have negative impacts on mosquito survival. Those have been extensively studied considering their application for biological control. It is, however, relevant to consider microorganisms that would improve the survival of microorganisms or even protect them against pathogens as they might influence the female oviposition choice and decrease the efficiency of pathogens in the field.

## 4. Did Mosquitoes and Microorganisms Evolve toward Traits Influencing the Behavior?

It is important to point out that not all microorganisms-induced behavioral manipulations of mosquitoes are adaptive. Influence of microorganisms on macroorganisms behavior has often been investigated in host-parasite systems and several points (based on the work of Poulin and Tinbergen) should be addressed to interpret those interactions as potential (co-)evolutionary output [99]. Those should help to understand if modifications of gravid females’ behavior lead to higher fitness benefits for each partners and if they evolve according to shifts in selective pressure. The four criteria or questions to address are (i) the fitness output of the interaction—What is it for?; (ii) the causation or mechanism—How does it work?; (iii) the ontology—How does it develop during the mosquito/microorganism lifetimes?; and (iv) the evolution—How did it evolve over the evolutionary history of both partners? Such questions are still far from being answered. Most efforts focused on the second question by describing microorganisms synthesizing semiochemicals that influence gravid female’s behavior. The first question was mostly investigated from the mosquito’s perspective by highlighting the consequences of such interactions on the trait of the progeny. Studies on other insects model often report manipulation of behavior that favors dispersal, survival, and growth of microorganisms opening a gate for future microorganisms centered studies [100]. Further investigations are mandatory to answer both the first and the second question with the same organisms and create some connectivity between the modification of behavior, its mechanism, and its fitness benefit. To our knowledge, the third and the fourth questions have not been addressed yet and would probably be essential to determine the specificity of such mosquito trait modification and their convergence or divergence across different lineage of mosquitoes and microorganisms.

## 5. Conclusions

There is a growing number of evidences showing how environmental microorganisms might be involved in behavioral modification of gravid females with major influence on the resulting oviposition decisions. Considering that microorganisms are also influencing the performance of offspring with some carry over effects on adults, it seems relevant to further investigate the evolutionary forces that drive the gravid female behaviors. However, several black boxes still remain to be opened. Firstly, in many cases the mechanisms through which microorganisms impact the female oviposition have not been fully deciphered. Secondly, there is a need for further studies on the stimulating and deterrent effects of microorganisms. Thirdly, the influence of microorganisms on oviposition behavior, progeny performance, and their transgenerational consequences have never been investigated in a single study to our knowledge. All those studies should also be convoluted in the future by integrating spatiotemporal dynamics of microbial communities in water containers. Further investigations following those directions are mandatory to develop more specific, environmentally friendly, and efficient strategies for biocontrol of mosquito vectors.

## Figures and Tables

**Figure 1 microorganisms-09-01589-f001:**
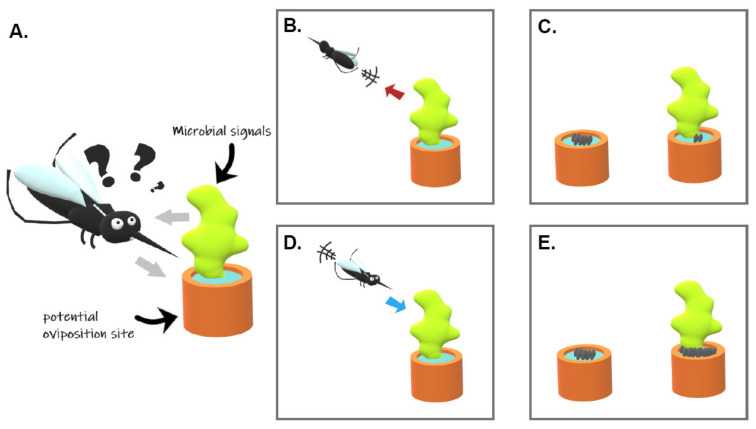
Behavioral responses of mosquitoes to microbial communities within breeding sites. Gravid female mosquitoes are able to (**A**) modify their behavior in response to visual, olfactive, gustative, or tactile cues that are directly or indirectly linked with the presence of microbial communities. The response can lead to a lower amount of eggs laid in the container whenever the cues are (**B**) repulsive (females will move away from the breeding site) or (**C**) deterrent (the production of eggs in the container will be reduced). On the opposite, the change in behavioral response can lead to a larger amount of eggs laid whenever the cues are (**D**) attractive (females will be oriented toward the breeding site), or (**E**) stimulant (the production of eggs in the container will be increased) (drawing: Minard G.).

**Table 1 microorganisms-09-01589-t001:** Microoganisms that influence the oviposition strategy of *Aedes aegypti, Ae. albopictus*, and *Anopheles gambiae* mosquitoes.

Microorganisms	Species	Condition/Concentration	Mosquito Species					Semiochemicals	References
			*Aedes aegypti*	*Aedes albopictus*	*Anopheles gambiae*	*An. arabiensis*	*Culex quinquefasciatus*		
Bacteria	*Bacillus thuringiensis*	10^6^ CFU/mL	attractivity/stimulation	no response	−	−	−		[49]
		10^7^ CFU/mL	attractivity/stimulation	attractivity/stimulation	−	−	−		
		10^8^ CFU/mL	no response	repellency/deterrence	−	−	−		
	*Bacillus thuringiensis* var. *israelensis*	0.5–2 mg/L (for Cx. quinquefasciatus), 8 mg/L (for *Ae. albopictus*), 2–6 mg/L (for *An. arabiensis)*	−	no response or attractivity/stimulation	−	no response	repellency/deterrence		[50,51,52,53]
	*Brevundimonas vesicularis*	10^6^ CFU/mL	attractivity/stimulation	attractivity/stimulation	−	−	−		[49]
		10^7^ CFU/mL	attractivity/stimulation	no response	−	−	−		
		10^8^ CFU/mL	no response	repellency/deterrence	−	−	−		
	*Citrobacter freundii*	10^6^ CFU/mL	attractivity/stimulation	no response	−	−	−		[49]
		10^7^ CFU/mL	attractivity/stimulation	attractivity/stimulation	−	−	−		
	*Comamonas spp*	[4.2 × 10^7^; 8.1 × 10^7^] CFU/mL	−	−	attractivity/stimulation	−	−	2-Methyl-3-decanol, methyl-1-butanol, 2-phenylethanol, phenylmethanol, alkyl-pyrazines, 3-methylbutanoic acid	[40]
	*Enterobacter asburiae*	[10^6^;10^7^] CFU/mL	attractivity/stimulation	no response	−	−	−		[49]
	*Enterobacter cancerogenus*	[10^6^;10^7^] CFU/mL	attractivity/stimulation	no response	−	−	−		[49]
	*Enterobacter gergoviae*	10^6^ CFU/mL	attractivity/stimulation	no response	−	−	−		[49]
		10^8^ CFU/mL	no response	repellency/deterrence	−	−	−		
	*Enterobacter ludwigii*	10^6^ CFU/mL	attractivity/stimulation	no response	−	−	−		[49]
		10^7^ CFU/mL	no response	attractivity/stimulation	−	−	−		
	*Exiguobacterium spp*	[5.2 × 10^7^; 5.3 × 10^7^] CFU/mL	−	−	attractivity/stimulation	-	-	2-Methyl-3-decanol, methyl-1-butanol, 2-phenylethanol, phenylmethanol, alkyl-pyrazines, 3-methylbutanoic acid	[40]
	*Lactococcus lactis*	10^6^ CFU/mL	attractivity/stimulation	no response	−	−	−		[49]
		10^7^ CFU/mL	attractivity/stimulation	attractivity/stimulation	−	−	−		
	*Micrococcus. spp*	[7.7 × 10^6^; 1.8 × 10^7^] CFU/mL	−	−	attractivity/stimulation	−	−	2-Methyl-3-decanol, methyl-1-butanol, 2-phenylethanol, phenylmethanol, alkyl-pyrazines, 3-methylbutanoic acid	[40]
	*Proteus spp*	[6.9 × 10^7^; 3.2 × 10^8^] CFU/mL	−	−	attractivity/stimulation	−	−	2-Methyl-3-decanol, methyl-1-butanol, 2-phenylethanol, phenylmethanol, alkyl-pyrazines, 3-methylbutanoic acid	[40]
	*Pseudomonas fulva*	10^7^ CFU/mL	attractivity/stimulation	no response	−	−	−		[49]
	*Pseudomonas plecoglossicida*	10^6^ CFU/mL	no response	repellency/deterrence	−	−	−		[49]
		10^7^ CFU/mL	no response	attractivity/stimulation	−	−	−		
	*Rhizobium huautlense*	10^8^ CFU/mL	repellency/deterrence	no response	−	−	−		[49]
	*Shigella dysenteriae*	[10^6^;10^7^] CFU/mL	attractivity/stimulation	no response	−	−	−		[49]
	*Vibrio metschnikovii*	[2 × 10^8^; 4 × 10^8^] CFU/mL	−	−	attractivity/stimulation	−	−	2-Methyl-3-decanol, methyl-1-butanol, 2-phenylethanol, phenylmethanol, alkyl-pyrazines, 3-methylbutanoic acid	[40]
Fungi	*Fusarium fujikuroi* complex		−	−	attractivity/stimulation	−	−	Cedrol	[43]
	*Fusarium falciforme*		−	−	attractivity/stimulation	−	−		
	*Smittium morbosum*	infected larvae	repellency/deterrence	−	−	−	−		[48]
	*Candidatus* near *pseudoglaebosa*	infected larvae	attractivity/stimulation	−	−	−	−		
	*Edhazardia aedis*		repellency/deterrence	−	−	−	−		[54]
Protist	*Ascogregarina taiwanensis*	infected larvae (12–97 trophozoites)	attractivity/stimulation	−	−	−	−		[48]
Trematode	*Plagiorchis elegans*	infected larvae	repellency/deterrence	−	−	−	−		[55,56]

“Attractivity” means that microorganisms elicit insect-oriented movement toward the source; “stimulation” means that microorganisms elicit oviposition; “repellency” means that microorganisms induce insect-oriented movement away from the source; “deterrence” means that microorganisms prevent oviposition.

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
