# Peer review of "Microorganisms Associated with Mosquito Oviposition Sites: Implications for Habitat Selection and Insect Life Histories"

_microorganisms, 2021, doi:10.3390/microorganisms9081589_

Round 1
Reviewer 1 Report
Girard et al review the influence that microorganisms in the water might have on mosquito oviposition, as well as how these microorganisms can affect the progeny's fitness. This topic is important as some of these microbes could be use as vector control or at least as deterrent for mosquito breeding sites.
I enjoyed reading this review. It is well written and it seems complete. The topic is very interesting and shows a novel approach to mosquito control.
The manuscript does not need many changes.
Author Response
Reviewer 1 :
Girard et al review the influence that microorganisms in the water might have on mosquito oviposition, as well as how these microorganisms can affect the progeny's fitness. This topic is important as some of these microbes could be use as vector control or at least as deterrent for mosquito breeding sites.
I enjoyed reading this review. It is well written and it seems complete. The topic is very interesting and shows a novel approach to mosquito control.
The manuscript does not need many changes.
[ANSWER] We would like to thanks the reviewer for her/his positive comments.
Reviewer 2 Report
The authors have done a rigorous review on the effects of different microorganisms and their metabolic secretions at mosquito oviposition sites. In my opinion, this review is very well written and covered a major area of mosquito habitat and behavioral insights. Some very important impacts have been highlighted in this review that have potential implications on biological mosquito control. The involvement of the microorganisms has been widely covered in the breeding sites, at different phases of mosquito development starting from egg hatching. The authors have also mentioned the insight of evolutionary forces in this this review. Although they have not discussed this area elaborately. I would encourage to do if it is possible. In that sense this review will be a very significant piece to read in this field.
I have few comments:
- The evolutionary aspects for both microorganisms and mosquitoes need more elaboration.
- The authors have tabulated the list of microorganisms and their direct/indirect effects on different mosquito species in Table 1. Sometimes these microorganisms do regulate mosquito activities and life cycle via different organic and inorganic chemical compounds. I wish to see another column in the same table or a separate table for these compounds and their effects. That would definitely ease the readers’ flow.
- Please strike out the text in () and keep the reference only in line 135.
- In Figure 1 legend, there is a typo. It should be (C) instead of (B) in line 506.
Author Response
Reviewer 2 :
The authors have done a rigorous review on the effects of different microorganisms and their metabolic secretions at mosquito oviposition sites. In my opinion, this review is very well written and covered a major area of mosquito habitat and behavioral insights. Some very important impacts have been highlighted in this review that have potential implications on biological mosquito control. The involvement of the microorganisms has been widely covered in the breeding sites, at different phases of mosquito development starting from egg hatching. The authors have also mentioned the insight of evolutionary forces in this this review. Although they have not discussed this area elaborately. I would encourage to do if it is possible. In that sense this review will be a very significant piece to read in this field.
I have few comments:
- The evolutionary aspects for both microorganisms and mosquitoes need more elaboration.
[ANSWER] We thank the reviewer for her/his positive comments and for her/his suggestion on the evolutionary forces driving those interactions. We did not develop this point in too much detail within the first version of our manuscript due to the lack of evidence on such adaptive modification of mosquitoes’ behavior by environmental microbes. However, we acknowledge that this point is of main interest for future prospect. We decided to add a paragraph with some theoretical concepts on the influence of microbes on macroorganisms behavior and pointed out the remaining pieces of the puzzle in order to help our research community to focus on those promising aspects. This paragraph can be found lines 500 to 524.
- The authors have tabulated the list of microorganisms and their direct/indirect effects on different mosquito species in Table 1. Sometimes these microorganisms do regulate mosquito activities and life cycle via different organic and inorganic chemical compounds. I wish to see another column in the same table or a separate table for these compounds and their effects. That would definitely ease the readers’ flow.
[ANSWER] We agreed and updated our Table 1 with an extra-column describing the semiochemicals (when identified).
- Please strike out the text in () and keep the reference only in line 135.
- In Figure 1 legend, there is a typo. It should be (C) instead of (B) in line 506.
[ANSWER] The points 3 and 4 were addressed accordingly.